

# Infrared and visible image fusion algorithm based on gradient attention residuals dense block

Yongyu Luo[1,2] and Zhongqiang Luo[1,2]

[1] School of Automation and Information Engineering, Sichuan University of Science and Engineering, Yibin, Sichuan, China
[2] Artificial Intelligence Key Laboratory of Sichuan Province, Sichuan University of Science and Engineering, Yibin, Sichuan, China

## ABSTRACT

The purpose of infrared and visible image fusion is to obtain an image that includes both infrared target and visible information. However, among the existing infrared and visible image fusion methods, some of them give priority to the fusion effect, often with complex design, ignoring the influence of attention mechanisms on deep features, resulting in the lack of visible light texture information in the fusion image. To solve these problems, an infrared and visible image fusion method based on dense gradient attention residuals is proposed in this article. Firstly, squeeze-and-excitation networks are integrated into the gradient convolutional dense block, and a new gradient attention residual dense block is designed to enhance the ability of the network to extract important information. In order to retain more original image information, the feature gradient attention module is introduced to enhance the ability of detail information retention. In the fusion layer, an adaptive weighted energy attention network based on an energy fusion strategy is used to further preserve the infrared and visible details. Through the experimental comparison on the TNO dataset, our method has excellent performance on several evaluation indicators. Specifically, in the indexes of average gradient (AG), information entropy (EN), spatial frequency (SF), mutual information (MI) and standard deviation (SD), our method reached 6.90, 7.46, 17.30, 2.62 and 54.99, respectively, which increased by 37.31%, 6.55%, 32.01%, 8.16%, and 10.01% compared with the other five commonly used methods. These results demonstrate the effectiveness and superiority of our method.

# INTRODUCTION

Image fusion is an enhancement technology that combines the advantages of two kinds of images taken by different types of sensors in different environments so that images with more comprehensive scene representation information can be obtained (*Chen et al., 2021*). Infrared and visible image fusion is a widely used category. Visible light images have rich texture details and high resolution, showing a high sensitivity to lighting conditions. However, the background imaging environment often affects the visible light image sensor,

Corresponding author
Zhongqiang Luo,
luozhongqiang@suse.edu.cn

which leads to the blur of the target in-formation in the image. Infrared image has better target detection and recognition ability, which is obtained by infrared sensors relying on thermal radiation. However, infrared images are noisy and lack texture details (*Karim et al., 2023*). Infrared and visible light image fusion is to integrate the advantage information of these two modes together to generate a fusion image with both the texture in-formation of the visible light image and the significant target information of the infrared image (*Chen et al., 2023*). Infrared and visible image fusion is of vital application value in the subsequent semantic segmentation (*Ha et al., 2017*), night vehicle target detection (*Gao et al., 2022*) and other advanced visual tasks.

Due to the importance of infrared and visible image fusion, numerous researchers have proposed a lot of fusion methods in recent years and have made a lot of contributions. These approaches can be broadly divided into traditional approaches and deep learning-based approaches. Traditional methods usually use mathematical algorithms or filters for multi-scale decomposition and then design corresponding fusion rules according to their characteristics (*Yang, Yan & Wang, 2024*). Generally, there are five methods: the method based on multi-scale transformation (*Liu et al., 2014*; *Liu, Mei & Du, 2017*; *Zhang & Maldague, 2016*; *Chen et al., 2020*), the method based on sparse representation (*Li, Wu & Kittler, 2020*; *Liu et al., 2016*), the method based on subspace (*Cvejic, Bull & Canagarajah, 2007*; *Mou, Gao & Song, 2013*; *Fu et al., 2016*), the method based on compressed sensing (*Wojtaszczyk, 2010*; *Liu, Luo & Li, 2014*; *He et al., 2014*) and the mixed method (*Ma et al., 2017*; *Wang et al., 2020*; *Zhou et al., 2016*). However, traditional methods rely on hand-designed fusion rules, and their performance is greatly limited when dealing with some more complex scene fusion tasks. In recent years, more and more deep learning methods have been introduced into the field of image fusion. Due to its strong ability in image features, it has shown better performance than traditional methods (*Chen et al., 2023*). Methods based on deep learning can be roughly divided into the following three categories: convolutional neural network-based methods (*Zhang et al., 2020b*; *Xu et al., 2020b*; *Ma et al., 2021*), generative adversarial network-based methods (*Ma et al., 2019*, *2020b*, *2020a*; *Zhang et al., 2021*), and autoencoder-based methods (*Li, Wu & Kittler, 2021*; *Li & Wu, 2018*; *Li, Wu & Durrani, 2020*; *Hong, Wu & Xu, 2022*).

Residual joins are an important technique in deep learning methods (*He et al., 2016*; *Zhang & Demiris, 2023*). *Li, Wu & Durrani (2019)* introduced it into the field of infrared and visible image fusion in 2019. Many of the subsequent methods have shown excellent performance. For example, *Long et al. (2021)* proposed an unsupervised infrared and visible image fusion method (RXDNFuse) based on aggregated residual dense networks, combining the advantages of ResNet and DenseNet to overcome the limitations of artificially designed activity level measurements and fusion rules. Similarly, since 2019, *Li & Wu (2018)* introduced the dense connection block into the field of infrared and visible image fusion, and it has also been well applied. For example, *Wang et al. (2021)* proposed a unified multi-scale dense connection fusion network, UNFusion, which uses dense jump connections in encoder and decoder networks to effectively extract and reconstruct multi-scale depth features. *Yang & Zeng (2022)* proposed and designed an infrared and visible image fusion algorithm based on unsupervised dense networks—TPFusion. By

training two densely connected networks, texture information and source images are fused, respectively, thereby preserving more texture details. *Tang, Yuan & Ma (2022)* proposed SeAFusion, a semantic sensing real-time infrared and visible image fusion net-work, and designed a gradient residual dense block to enhance the network's perception of fine-grained spatial details. *Zuyan, Bin & Chang (2023)* proposed RDCa-Net, a residual dense channel attention symmetric network. A symmetric skipping attention network is constructed, in which the skipping attention mechanism can compensate for the information loss in the feature extraction stage. A weight block is also designed to calculate the weight of information in the loss function and adaptively retain the source image information. *Wang et al. (2024)* combined the advantages of ResNeXt and DenseNet and designed a new gradient-aggregated residual dense block (GRXDB) algorithm that can maintain both strong texture features and weak texture features. At the same time, the spatial and channel attention mechanism is introduced to refine the channel and spatial information of the feature map and enhance the information capture ability of the feature map. *Boroujeni & Razi (2024)* realized the con-version from RGB to IR images through an improved conditional generation adversarial network and applied it to forest fire monitoring.

Although these deep learning methods surpass the traditional fusion methods in processing speed and fusion effect, they all have the problem of too many parameters to some extent. Dense networks will add previous features to the next operation, resulting in the network system being too large. Most of the methods blindly pursue the fusion effect, introduce too complex structure, and ignore the extraction of deeper features. As a result, the final fusion image lacks some relevant information and also brings a certain degree of noise interference, which ultimately leads to the decline of the quality of the fusion image. To solve these problems, *Zou et al. (2023)* proposed a lightweight infrared and visible image fusion network based on edge-guided dual attention and designed a feature gradient attention block to reduce network parameters as much as possible without affecting feature extraction. The advantages and disadvantages of the above-mentioned methods are shown in Table 1.

In general, in the process of fusion of infrared image and visible image, we need to avoid introducing too complex structure while ensuring the quality of the final fusion image. In this article, a fusion method of infrared and visible images based on gradient attention residuals dense block is proposed. Firstly, the feature gradient attention module is used to extract the feature from the source image, and then it is sent to the gradient attention residual dense block to enrich the edge feature information. The extraction of deep features is enhanced effectively, while the network parameters are reduced and the fusion efficiency is improved. The fusion layer uses an energy attention mechanism, which can strengthen the weight of the target region. The fusion layer and the feature extraction layer form a complementary association, which greatly improves the fusion quality. Finally, the fusion image is reconstructed through four convolutions.

The main contributions of our algorithm are as follows:

**Table 1 Summary of advantages and disadvantages of various image fusion methods.**

| Method name | Advantages | Disadvantages |
| --- | --- | --- |
| ResNet and Zero-Phase Component Analysis | Using ResNet to extract depth features and combining with ZCA to normalize features, better evaluation performance is obtained. | Direct use of depth features may result in degradation of fusion performance in some cases. |
| RXDNFuse | Combining the structural advantages of ResNeXt and DenseNet, it overcomes the limitations of manual design and automatically estimates the degree of information preservation. | The design is complicated, the parameters are too many, and the training time is long. |
| DenseFuse | The coding network combines the convolutional neural network layer, fusion layer and dense block to preserve the depth feature and improve the fusion effect. | Network parameters need to be further reduced to improve efficiency. |
| UNFusion | In the encoder and decoder network, dense jump connections are used to extract and reconstruct multi-scale depth features effectively. | High computational complexity. |
| TPFusion | More texture details are preserved by training two densely connected networks to fuse texture information and source images, respectively. | Large computing resources are required. |
| Semantic-aware Real-time Fusion Network | Gradient residual dense blocks are designed to enhance the perception of fine-grained spatial details. | Processing speed is limited. |
| Lightweight Edge-guided Dual Attention | A feature gradient attention block is designed to reduce network parameters while maintaining feature extraction. | In some complex scenarios, the details remain insufficient. |
| RDCa-Net | A symmetric skip attention network is constructed, and a skip attention mechanism is used to compensate for the information loss in the feature extraction stage. | Its performance in different scenarios needs to be further verified. |
| SCGRFuse | Improved fusion image quality based on spatial/channel focus mechanism and gradient aggregation of residual dense blocks. | Its stability and generalization ability need to be further verified. |

- Proposes a residual-intensive module based on gradient attention. The squeeze-and-excitation networks introduced in this module can preserve more details and enhance the sensitivity of the network to fine-grained features.
- In the feature extraction part, a depth-separable convolution-based gradient feature attention module is introduced. By combining this module with gradient attention residual dense block, the edge feature information of the final fused image is greatly enriched.
- Our algorithm can extract more information and retain more edge details than other algorithms. The problem of too many parameters in the deep learning method is solved. On the open data set, both subjective and objective evaluation have better performance.

The rest of this article is organized as follows: 'Related Works' covers the modules related to our proposed method. 'Proposed Fusion Method' details the network model and the loss function. 'Experiment' presents experimental comparisons between our method and others, along with an ablation study. 'Future Works' provides a summary.

# RELATED WORKS
## Squeeze-and-excitation networks
The squeeze-and-excitation (SE) network is an architecture proposed by *Hu, Shen & Sun (2018)* to improve the representation capability of convolutional neural networks by

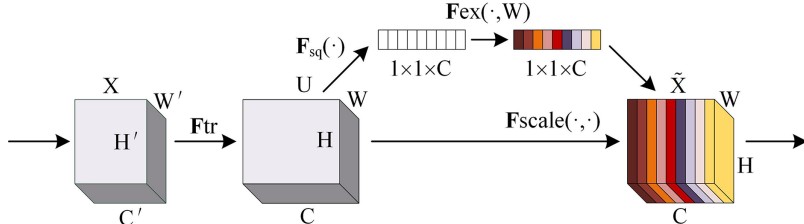

**Figure 1** SE network architecture diagram.

explicitly modeling the dependency relationship between channels and has achieved remarkable results in multiple tasks in the field of computer vision. SE network is through automatic learning with another new neural network to obtain the importance of each channel of the feature map and use it to assign a weight to each feature so as to achieve the effect of making the neural network focus on certain feature channels. Feature graph channels that are useful to the current task can be promoted, while those that are not useful can be suppressed. The SE network architecture diagram is shown in Fig. 1.

SE network implementation steps are as follows:

(1) Squeeze: The spatial information of each channel is compressed into an A scalar through global average pooling, and the feature map is moved from $[h, w, c]$ to $[1, 1, c]$ to capture the global spatial information.

(2) Excitation: The weight of each channel is generated through two fully connected layers and a nonlinear activation function. First, a fully connected layer is used to reduce the number of channels to 1/16 of the original, and then nonlinearity is introduced through the ReLU activation function. A fully connected layer is then used to restore the number of channels to their original size, and the weight of each channel is generated by the sigmoid function $[1, 1, c] \Rightarrow [1, 1, c]$.

(3) Recalibration: The resulting channel weight is weighted to the original feature, thereby recalibrating the feature response of each channel. Enables the network to adaptively enhance important features $[h, w, c] * [1, 1, c] \Rightarrow [1, 1, c]$.

The SE network is not judged directly according to the numerical distribution of feature channels, but its core idea is to automatically learn feature weights according to loss through the fully connected network so that the weight of effective feature channels is greater. With their simple structure and easy integration into existing convolutional neural networks, SE modules can bring significant performance improvements to existing state-of-the-art deep architectures at minimal additional computational cost. The SE module can improve the capability of feature representation by adaptive re-calibration of channel feature response. This adaptability makes the SE module more flexible in dealing with different image fusion tasks. The SE module has a low computational complexity compared to other attention modules, which makes it more efficient in practical applications.

## Fusion of local energy features

Local energy feature fusion is a traditional image fusion strategy. The fusion method will first calculate the energy of the local region in the infrared image and visible image, and

then judge the energy size of the same region in the two images, and finally calculate the weight according to the energy size, and then fuse the pixel of the point. The definition of local energy characteristics is shown in Formula (1).

$$S(i,j) = \sum_{m}^{i} \sum_{n}^{j} C(i+m, j+n)^2 \tag{1}$$

where $S(i,j)$ is the position of image $S$ at the pixel point $(i,j)$. The sum of the squares of the pixels in the $m \times n$ window centered on the pixel point is the local energy of the point. In the fusion operation, the matching degree between the same area of two images is calculated first, and then it is used as the fusion criterion. The calculation of matching degree is shown in Formula (2).

$$M_{AB} = \frac{\left( \sum_{m}^{i} \sum_{n}^{j} C_A(i+m, j+n) \cdot C_B(i+m, j+n) \right)^2}{S_A(i,j) \cdot S_B(i,j)} \tag{2}$$

where $M$ represents the matching degree of energy in the same region between image $A$ and image $B$. $C$ represents the local energy of pixel $(i,j)$.

Set the threshold to $e$; if $M < e$, select the pixel with the larger local energy and discard the remaining pixels; if $M > e$, then the weights of pixels with small local energy are shown in Formula (3).

$$W_{\min} = 0.5 \times \left( 1 - \frac{1 - M_{AB}}{1 - e} \right) \tag{3}$$

The weights of pixels with large local energy are shown in Formula (4).

$$W_{\max} = 1 - W_{\min} \tag{4}$$

This traditional local energy fusion algorithm artificially sets the threshold and adopts a fixed weight formula, which will cause shortcomings such as image artifacts and is only applicable to pixel-level fusion (*Zou et al., 2023*).

## PROPOSED FUSION METHOD

In this part, we give a comprehensive description of the proposed network and then introduce the image feature extraction module, energy fusion module, and loss function in detail.

### Overall network structure

The network structure diagram of the overall model is shown in Fig. 2. Our model consists of three parts: feature extraction module, fusion module, and feature reconstruction module.

The infrared and visible images $F^l$ of the test are given (for infrared images $l = ir$, for visible images $l = vis$). The initial feature $\Psi^l$ is represented by Formula (5).

$$\Psi^l_{\text{extract1}} = F_{FGAB}(F^l) \tag{5}$$
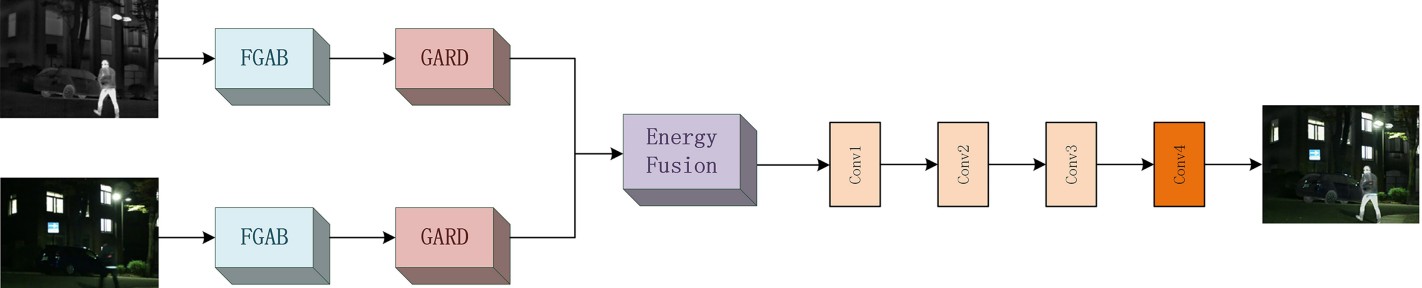

**Figure 2 Overall network structure diagram.**

where $F_{FGAB}$ stands for the feature gradient attention block (FGAB). After passing through the module, the features of the infrared and visible images will be extracted for the first time. In order to extract more detailed information, the overall fusion effect is improved. We further use a gradient attention residual dense block for information extraction. The global feature $\Psi^l_{extract2}$ obtained is represented by Formula (6).

$$\Psi^l_{extract2} = F_{LGRDB}(\Psi^l_{extract1}) \tag{6}$$

where stands for gradient attention residual dense block (GARD). The infrared and visible image features will be extracted in depth a second time in the module to ensure more complete global information. Then, we use the energy fusion method to fuse the two information and obtain the global feature after fusion, which can be expressed by Formula (7).

$$\Psi_F = F_{EN}(\Psi^l_{extract2}) \tag{7}$$

where $F_{EN}$ stands for energy fusion operation. Finally, the image is reconstructed to obtain the final fusion image $I_F$, which can be expressed by Formula (8).

$$I_F = F_{re}(\Psi_F) \tag{8}$$

where $F_{re}$ represents the image reconstruction operation. The reconstruction module consists of three 3 * 3 convolution layers F and one 1 * 1 convolution layer. The 3 * 3 convolutional layer uses leakage rectified linear element as the activation function, and the 1 * 1 convolutional layer uses Than as the activation function (The specific parameter settings are shown in Table 2).

## Image feature extraction module

The feature extraction module is the key part of infrared and visible image fusion. In this part, we use the feature gradient attention module (FGAB) and the gradient attention residual intensive module (GARD) to extract features of infrared and visible images, respectively. Below we will explain these two modules in detail.

The internal structure diagram of FGAB is shown in Fig. 3. FGAB module follows the idea of gradient residual-dense block. In the mainstream, a 1 * 1 convolution and a 3 * 3 convolution are first used to extract basic features. In residual flow, the Sobel operator is

| Table 2 Network configuration of the convolutional layers. | | | |
|---|---|---|---|
| Layer | Kernel size | Stride | Padding |
| Conv1 | 3 | 1 | 1 |
| Conv2 | 3 | 1 | 1 |
| Conv3 | 3 | 1 | 1 |
| Conv4 | 1 | 1 | 1 |

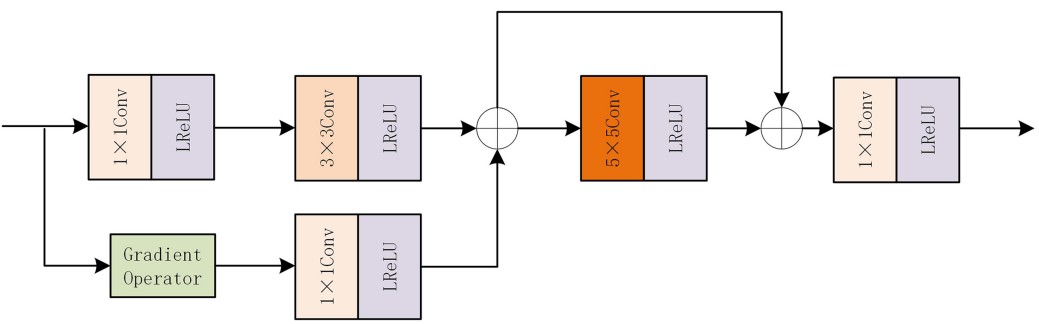

**Figure 3 FGAB internal structure diagram.**

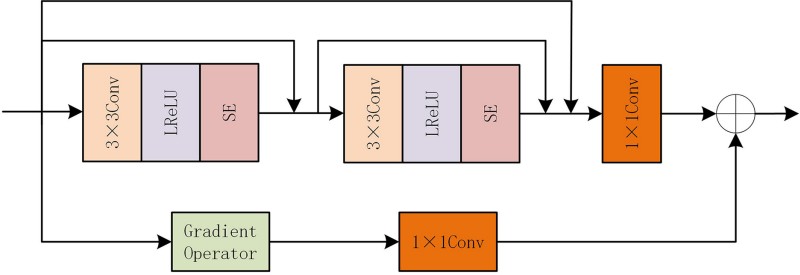

**Figure 4 GARD internal structure diagram.**

used as a gradient operator to extract the detailed features, and then the gradient information is optimized by 1 * 1 convolution. The information in the main stream and residual stream is added, the feature extraction is enhanced by grouping convolution, and the original feature is preserved by skip connection. The element addition strategy is then used to increase the weight of local features. The final 1 * 1 convolution is used to increase the interactivity of the channel information.

After the feature extraction module of FGAB, although the global information of the image has been extracted, the detailed information is not enough. So we designed a GARD module. The internal structure of GARD is shown in Fig. 4. Also following the idea of gradient residual dense blocks, the mainstream uses dense connections, which are composed of two 3 * 3 LReLU convolutions and one 1 * 1 common convolutional layer. We add SE attention modules at the end of the two convolutional layers, respectively, to improve the network's perception of important information. The residual flow is combined with the gradient operation to calculate the gradient size, and the 1 * 1

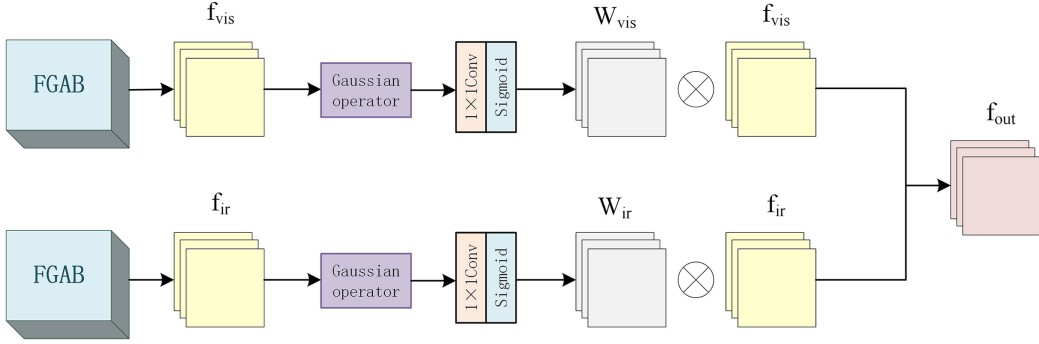

**Figure 5 Fusion layer internal structure diagram.**

convolution layer is used to eliminate the channel dimension difference. Finally, the depth feature is combined with the fine-grained feature by the element addition method.

## Fusion layer

In the fusion layer part, different from the above traditional local energy fusion algorithm, this is an energy fusion structure based on Gaussian operators, as shown in Fig. 5.

Since the energy size of each pixel is related to its surrounding pixels, the weight of the surrounding pixels is inversely proportional to its distance from the center pixel. So, add a 3 * 3 Gaussian operator with $\sigma = 0.8$ to calculate the local energy. The features of infrared image and visible image are represented by $f_{ir}$ and $f_{vis}$, respectively, and the local energy $S_{ir}(i,j)$ and $S_{vis}(i,j)$ of each pixel calculated by the Gaussian operator are shown in Formulas (9) and (10) respectively.

$$S_{ir}(i,j) = \sum_{m}^{i}\sum_{n}^{j} G_{ir}(i+m,j+n)C_{ir}(i+m,j+n) \tag{9}$$

$$S_{vis}(i,j) = \sum_{m}^{i}\sum_{n}^{j} G_{vis}(i+m,j+n)C_{vis}(i+m,j+n) \tag{10}$$

where $G$ represents the Gaussian operator and $C$ represents the region where the feature map is computed.

Then, the energy weights of each region of the infrared and visible images are calculated by the convolutional layer with the kernel size of 1 * 1 and the Sigmoid activation function, and the corresponding energy weight graphs are generated. The process is shown in Formulas (11) and (12).

$$W_{ir} = conv\_weight_{ir}(f_{ir}^{1}) \tag{11}$$
$$W_{vis} = conv\_weight_{vis}(f_{vis}^{1}) \tag{12}$$

where $f_{ir}^{1}$ and $f_{vis}^{1}$ represent infrared and visible image features processed by the above Gaussian operation.

Then, the original features $f_{ir}$ and $f_{vis}$ are multiplied element by element with their corresponding weights $W_{ir}$ and $W_{vis}$, respectively, and then added with the original feature to obtain features $f_{ir}^{2}$ and $f_{vis}^{2}$, as shown in Formulas (13) and (14).

$$f_{ir}^2 = f_{ir} \odot W_{ir} + f_{ir} \tag{13}$$
$$f_{vis}^2 = f_{vis} \odot W_{vis} + f_{vis} \tag{14}$$

Finally, the processed features $f_{ir}^2$ and $f_{vis}^2$ are spliced to obtain the final output $f_{out}$, as shown in Formula (15).

$$f_{out} = concat(f_{ir}^2, f_{vis}^2) \tag{15}$$

## Loss function

In order to retain the detail and intensity information of the original infrared image and visible image, intensity loss $\mathscr{L}_{int}$ and detail loss $\mathscr{L}_{details}$ are used as loss functions to train the network during the training stage. Intensity loss can constrain the pixel intensity of the fused image to be consistent with that of the source image as much as possible, and its definition is shown in Formula (16).

$$\mathscr{L}_{int} = \frac{1}{HW} \| I_f - \max(I_{vis}, I_{ir}) \|_2 \tag{16}$$

where $H$ and $W$ represent the height and width of the image, respectively; $\| \cdot \|_2$ stands for $l_2$ norm; $\max(\cdot)$ indicates the maximum element selection. The working principle is to select the maximum pixel intensity between the infrared image and the visible image and then constrain the brightness distribution of the infrared image and the visible image. But it can only reduce the difference in pixels between the fused image and the original image. Therefore, in order to retain more texture details, detail loss is required to maintain consistency between the fused image and the texture details of the source image. The definition of detail loss is shown in Formula (17).

$$\mathscr{L}_{details} = \frac{1}{HW} \| (\nabla I_f) - \max(|\nabla I_{vis}|, |\nabla I_{ir}|) \|_1 \tag{17}$$

where $\|\cdot\|_1$ represents $l_1$ norm; $\nabla$ stands for Sobel gradient operator; $|\cdot|$ is absolute value. The working principle is that the selection of the gradient maximum elements of infrared and visible images is a rule that restricts the texture details of fused images.

The total loss function $\mathscr{L}_{total}$ consists of strength loss $\mathscr{L}_{int}$ and detail loss $\mathscr{L}_{details}$, as shown in Formula (18).

$$\mathscr{L}_{total} = \mathscr{L}_{int} + \mathscr{L}_{details} \tag{18}$$

## EXPERIMENT

### Experimental settings

During the training phase, we trained our network using the MSRS (*Tang, Yuan & Ma, 2022*) dataset, which contains 1,444 pairs of high-quality aligned infrared and visible images, a new multispectral dataset of infrared and visible image fusion built on the MFNet dataset. The MSRS data set contains color images, and we need to convert the RGB visible image to YCrCb format before it can be fused with the single-channel infrared image. We first separated the Y channel containing the brightness information and fused it with the

infrared image, then spliced the fused image with the Cr channel and Cb channel, and finally converted it to the RGB color space.

Set the training parameters as follows: Set the batch size to 16, the epoch to 10, the initial learning rate to 30, and the learning rate for the second round of training to 16. The batch size of 16 is a balanced choice based on our hardware configuration and experimental results. Larger batch sizes can speed up training and reduce training time per epoch, but may result in training losses decreasing more slowly and possibly achieving higher minimum validation losses. The batch size of 16 not only ensures the convergence speed and stability of the model, but also ensures the efficiency of model training. The epoch is set to 30, based on the fact that the model has not significantly improved its performance on the validation set. More epochs help the model fully learn data features, but can also lead to overfitting. We verify losses by monitoring them and stop training when losses no longer decrease significantly to avoid overfitting. The setting of the initial learning rate is very important for the convergence rate and the final performance of the model. We chose a moderate initial learning rate to ensure that the model can converge quickly at the beginning of training, while reducing the learning rate to refine the model weight at the later stage of training to improve the stability and performance of the model. In the second round of training, the learning rate is reduced in order to make more fine-grained adjustments when approaching the optimal solution and avoid excessive step size jumping out of the optimal solution area.

This experiment is based on the Windows 10 operating system and the PyCharm software platform. The hardware configuration is a 36-core Intel(R) Xeon(R) CPU E5-2695 v4@2.10GHz processor, 160G memory, and an and an NVIDIA TITAN Xp graphics card with a total of 36G video memory. The experimental environment is Python 3.9.13, Cuda 11.3, and PyTorch 1.12.1.

## Comparative experiment

During the testing phase, we used the TNO (*Toet, 2017*) dataset to demonstrate the effectiveness of our network. The TNO dataset is a publicly available multi-band image dataset for the development and evaluation of image fusion algorithms. The dataset contains enhanced vision, near-infrared, and long-wave infrared images. We selected 42 pairs of infrared and visible images from the TNO dataset for exper-iments. We selected six commonly used deep learning methods: proportional maintenance of gradient and intensity (PMGI) (*Zhang et al., 2020a*), DenseFuse (*Li & Wu, 2018*), FusionGAN (*Ma et al., 2019*), RFN-Nest (*Li, Wu & Kittler, 2021*), swinfuse (*Wang et al., 2022*), and U2Fusion (*Xu et al., 2020a*) to process the images of the TNO dataset.

PMGI enhances the quality of fused images through multi-scale guided information injection. It provides a powerful benchmark to demonstrate the effectiveness of multi-scale feature fusion in image fusion. DenseFuse uses a dense network of connections to fuse images, a structure that helps capture richer contextual information. It represents an effective feature fusion strategy that enhances feature transfer through dense connections. FusionGAN is a generative adversarial network (GAN)-based fusion method that generates high-quality fusion images through adversarial training. FusionGAN was chosen

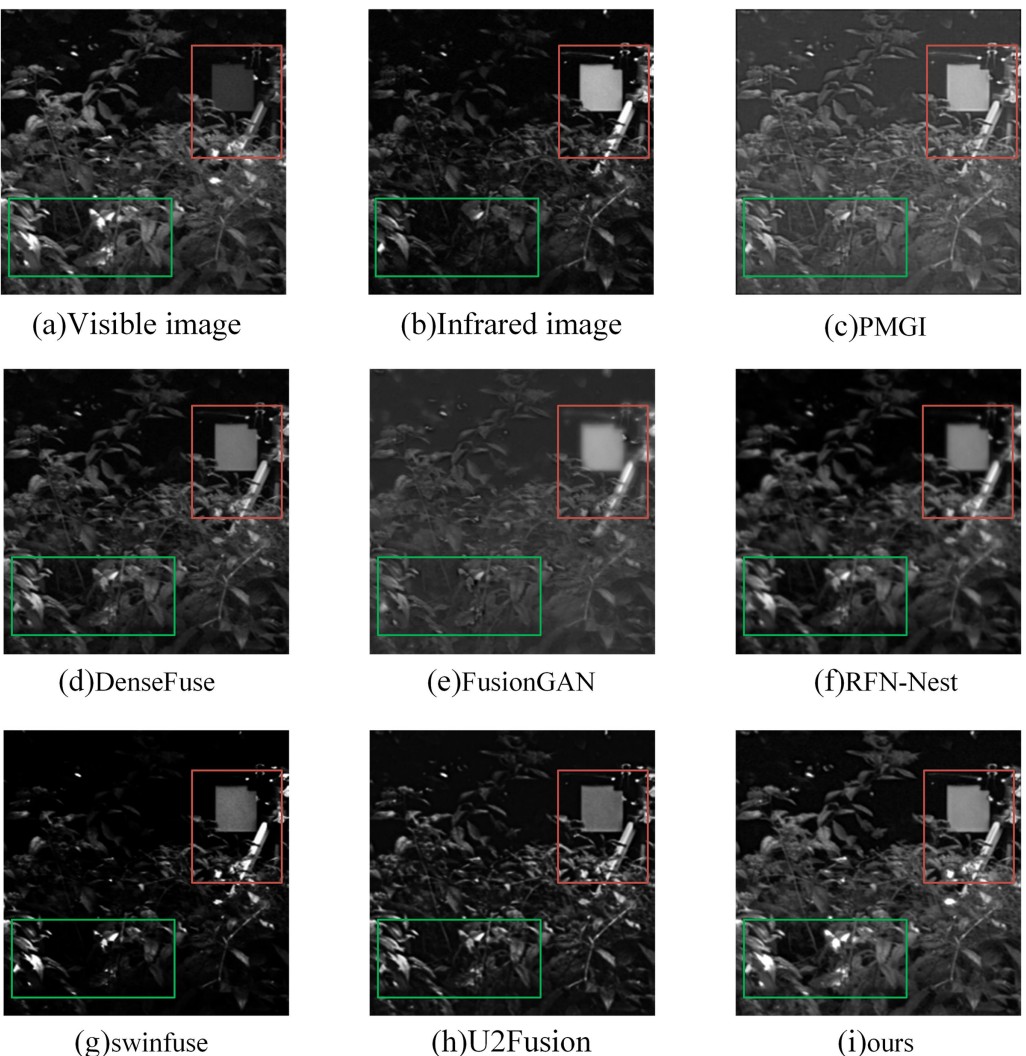

(a)Visible image     (b)Infrared image     (c)PMGI

(d)DenseFuse     (e)FusionGAN     (f)RFN-Nest

(g)swinfuse     (h)U2Fusion     (i)ours

**Figure 6 (A–I) Comparison of fusion results of various methods in a "forest" image.**

to demonstrate the potential and challenges of GAN-based approaches in image fusion. RFN-Nest improves fusion performance through residual learning and nested connections. RFN-Nest was chosen because it presents a novel network architecture that significantly improves feature extraction and fusion. SwinFusion uses Swin Transformer for image fusion, a new type of attention mechanism that captures global dependencies. We chose SwinFusion to demonstrate the advanced performance of transformer-based approaches in image fusion. U2Fusion is a unified unsupervised image fusion network that generates fusion results through feature extractors and information richness measurements. We chose U2Fusion because it provides an unsupervised learning perspective, which is very valuable for situations where labeled data is lacking in real-world applications.

The results obtained by these methods are compared with those proposed by us, and the fusion results of each method are analyzed qualitatively and quantitatively.

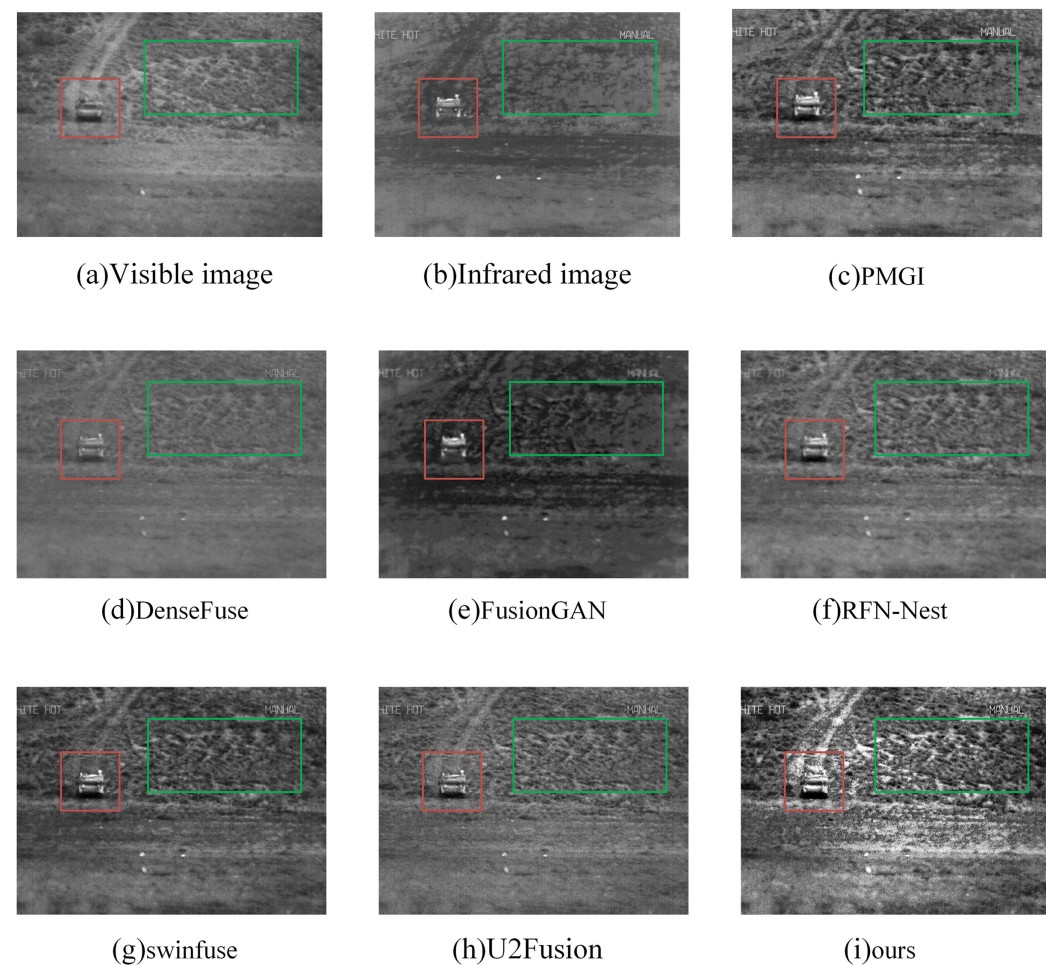

**Figure 7 (A–I) Comparison of fusion results of various methods in an "off-road vehicle" image.**

## Qualitative analysis

In the subjective evaluation, we selected three representative image pairs from 42 image pairs for observation and comparison, as shown in Figs. 6–8, where (a) is the visible image, (b) is the infrared image, (c) is the fusion image of the PMGI fusion method, (d) is the fusion image of the DenseFuse fusion method, and (d) is the fusion image of the Densefuse fusion method. (e) is the fusion image of the FusionGAN fusion method; (f) is the fusion image of the RFN-Nest fusion method; (g) is the fusion image of the swinfuse fusion method; (h) is the fusion image of the U2Fusion fusion method; (i) is the fusion image of our proposed fusion method.

Figure 6 is a comparison of the fusion results of various methods in the "forest" image. The objects in the red boxes in Figs. 6E and 6F appear to have more obvious boundary blurring, resulting in artifacts. At the same time, the details of leaf reflection in the green box in Fig. 6E are not obvious. The contrast of fusion results in Fig. 6C is obviously low. The visible light detail in Fig. 6D is rich, but the target in the red box is not obvious enough. In Fig. 6G, the target is more obvious in the red box, but the overall tone is

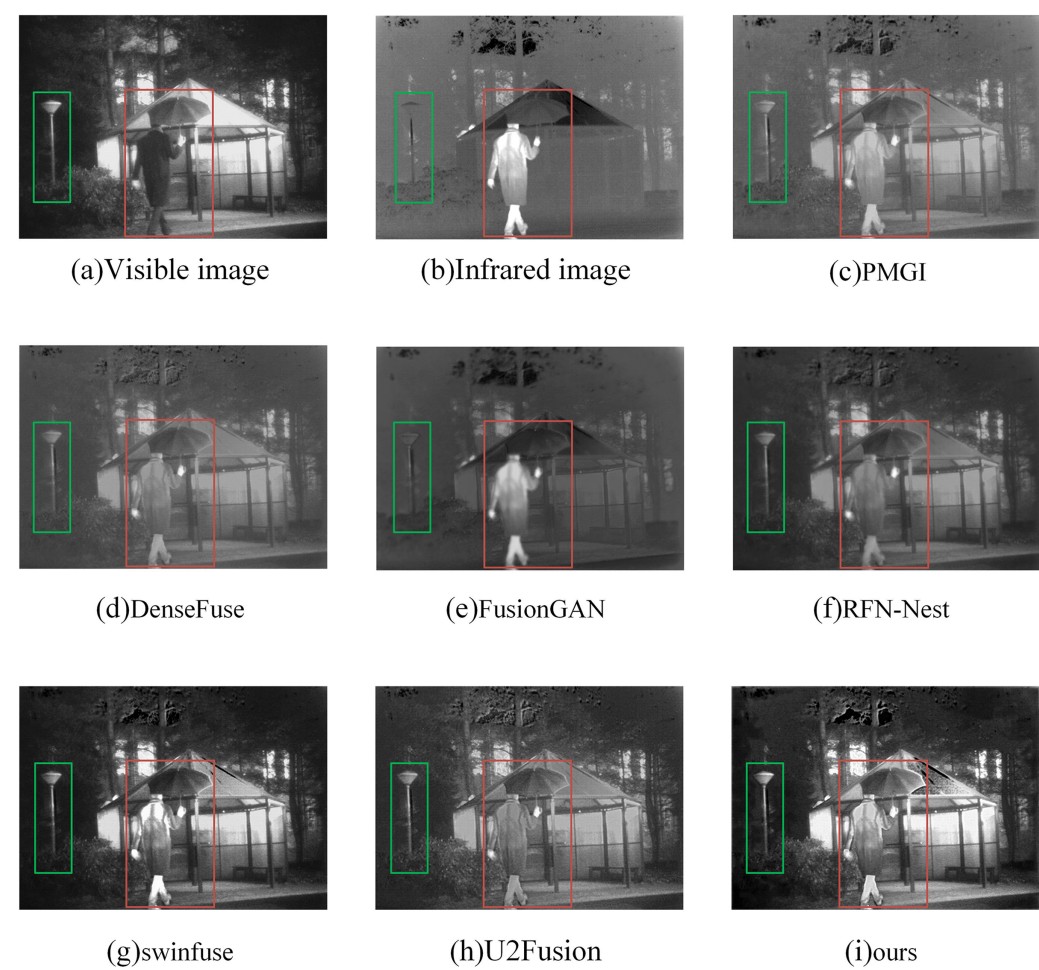

**Figure 8** **(A–I) Comparison of fusion results of various methods in a "human" image.**

dark, and the details of the leaves in the green box are seriously missing. The overall quality of Fig. 6H is good, but compared with our method, the detailed information of leaves in the green box is still lacking. Our method preserves the detailed information of the leaves in the green frame, and the infrared targets in the red frame are also very clear.

Figure 7 is a comparison of fusion results of various methods in the "off-road vehicle" image. The red box shows infrared target information, and the green box shows details. From the comparison of the results of these seven methods, we can see that the vehicle target in the red box of Figs. 7D and 7H is not obvious. In Fig. 7E and 7G, the vehicle target in the red box is more obvious, but in the green box, the detailed information of the lawn is more vague. This issue also arises in Figs. 7C and 7F. Our proposed method not only clearly shows the details of the lawn in the green box but also highlights the vehicle target in the red box.

Figure 8 is a comparison of the fusion results of various methods in "human" image. Through the observation of the "man holding an umbrella" in the red box of each figure,

Figs. 8D, 8F, and 8H, the infrared information of the man's back cannot be identified at a glance in these three pictures. In Fig. 8C, the figure is barely recognizable, but with reduced clarity. There are a lot of artifacts in Fig. 8E, and the street light information in the green box is almost invisible. The details in Fig. 8G are good in all aspects, but the details on the edge of the umbrella are still better than the method we proposed.

### Quantitative analysis

Because qualitative analysis is subjective and not comprehensive enough, quantitative analysis is needed. We used average gradient (AG) (*Yu et al., 2015*), information entropy (EN) (*Yonghong, 2012*), spatial frequency (SF) (*Li, Kwok & Wang, 2001*), mutual information (MI) (*Zhang et al., 2010*), standard deviation (SD) (*Wang & Chang, 2011*), fidelity of fused visual information (VIF) (*Han et al., 2013*), Quality Assessment Based on Fusion (Qabf) and the Sum of the Correlations of Differences (SCD) to analyze the experimental results. Figure 9 shows the results of these seven approaches on these eight metrics. Where the horizontal coordinate represents the number of test images, and the vertical coordinate represents the average value of the corresponding image evaluation index. The black lines represent the PMGI method, the purple lines represent the DenseFuse method, the green lines represent the FusionGAN method, the blue lines represent the RFN-Nest method, the yellow lines represent the swinfuse method, the cyan lines represent the U2Fusion method, and the red lines represent the method presented in this article.

As can be seen from the figure, compared with other methods, the method proposed in this article has achieved the best results in AG, EN, SF, MI, and SD. The results of SF, AG, and EN mean that the fusion image obtained by the proposed method has a higher resolution, reflecting more texture details in the source image. The optimal MI means that the proposed method has a better ability to transmit information according to the lighting conditions. The optimal result of SD verifies the validity of the model from the visual effect of the fused image. Although our approach is not in the leading position in VIF indicators, it still shows good performance. In the two indicators of Qabf and SCD, our method still needs to be improved, which will become the focus of our subsequent improvement research. In view of all the above indicators, our method has obvious advantages in maintaining global characteristics and detailed information and can achieve better fusion performance.

## Ablation experiment

### Gradient attention residual intensive modular analysis

In our method, the FGAB module extracts most of the global features of the source image, and on this basis, we also design the gradient attention residual intensive modular analysis (GARD) module to enhance the extraction of detailed information. The GARD module integrates the SE network into a gradient dense block (GRDB) to enhance the information-capturing ability of each channel. In order to verify the effectiveness of the added gradient dense fast and SE network, we conducted ablation experiments on 42 image pairs of the TNO dataset, all of which adopted the same parameter settings.

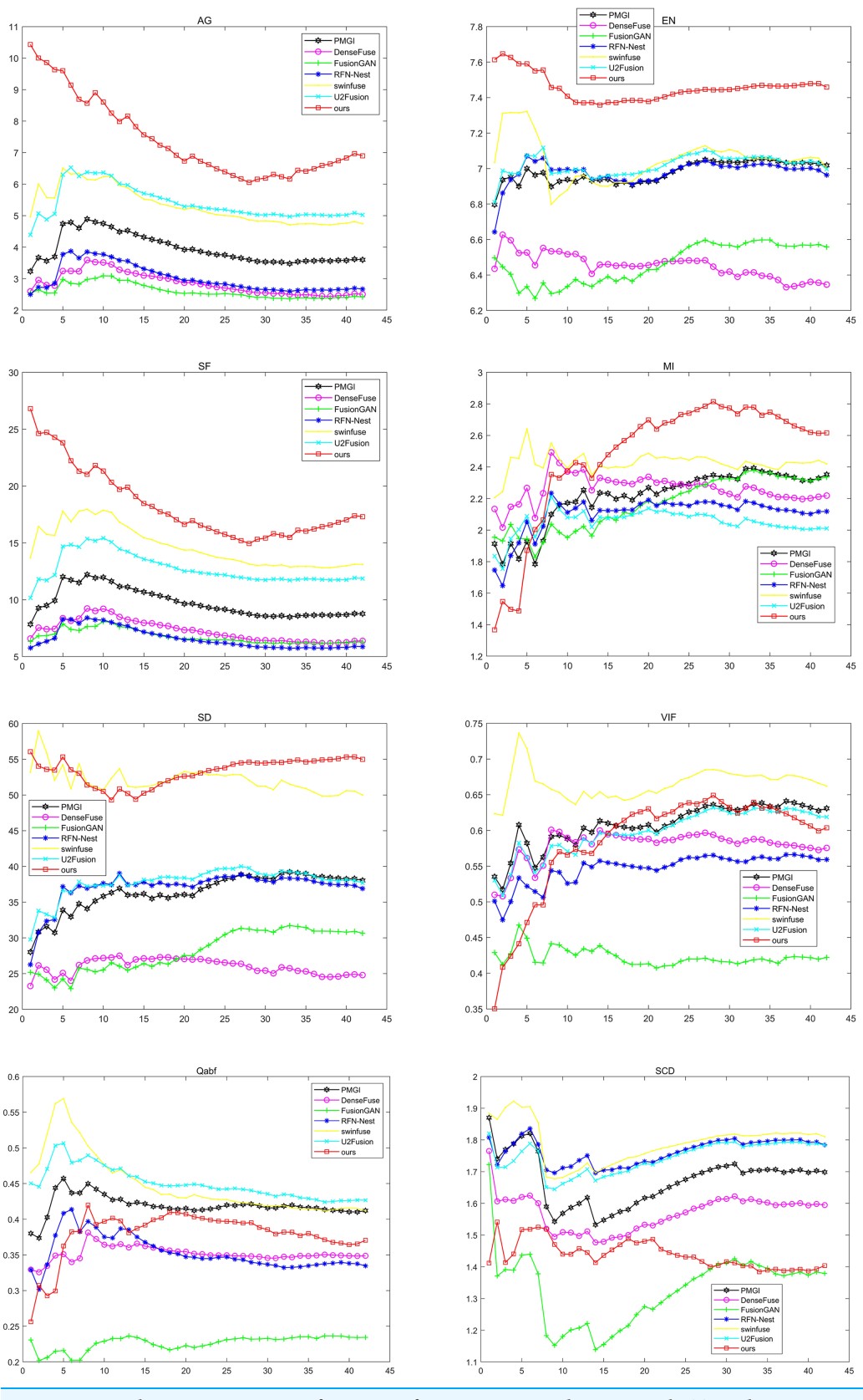

**Figure 9  Qualitative comparison of 42 pairs of images on six indicators on the TNO dataset.**

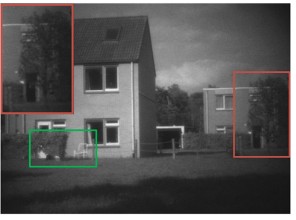
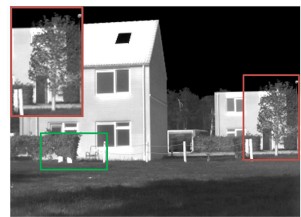

(a)Visible image        (b)Infrared image

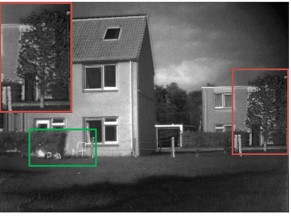
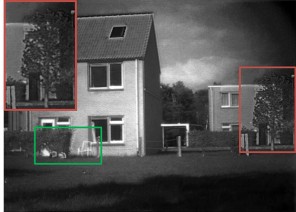
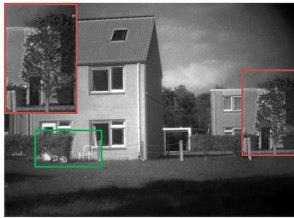

(c)No SE & GRDB      (d)No SE      (e)Ours

**Figure 10 (A–E) Experimental comparison of "house" image ablation.**

## Qualitative analysis

As shown in Fig. 10, we selected a set of images from the TNO data set for display and compared the local details magnified in the red box. No SE indicates the fusion result obtained after the SE network module is removed from the network. No SE & GRDB indicates the fusion result of removing the GARD module; ours represents the fusion result of the method proposed in this article. In the enlarged image at the top left, we can see that in the absence of the SE network, the infrared information of the house behind the trees is weaker and the brightness is significantly reduced. We continued to remove the SE network and residual gradient dense blocks when we found artifacts on the edge of the window and the edge of the leaves. This result shows that our GARD module can efficiently fuse image details while enhancing both infrared information extraction.

## Quantitative analysis

Seven indexes of mean gradient (AG) (*Yu et al., 2015*), information entropy (EN) (*Yonghong, 2012*), spatial frequency (SF) (*Li, Kwok & Wang, 2001*), standard deviation (SD) (*Wang & Chang, 2011*), fidelity of fused visual information (VIF) (*Han et al., 2013*), correlation coefficient (CC) (*Zhu & Bamler, 2012*), and sum of correlation differences (SCD) (*Aslantas & Bendes, 2015*) were selected to analyze the results of the ablation experiment. The experimental results are shown in Table 3, and the optimal results are shown in bold.

In the table, we can see more directly that among the seven indicators, five of the methods proposed by us have achieved optimal results. Among these two indicators, AG and SF, our proposed method shows greater advantages, which also means that the GARD module integrated with SE attention can extract more comprehensive and rich detailed information. This result also verifies the validity of our proposed method once again.

**Table 3  Ablation results of 42 pairs of images on the TNO dataset.** The best values are shown in bold.

| Method | SD | VIF | AG | CC | SCD | EN | SF |
|---|---|---|---|---|---|---|---|
| No SE & GRDB | 51.6671 | 0.6330 | 6.4277 | 0.4006 | 1.3372 | 7.3111 | 15.8626 |
| No SE | **56.6110** | **0.6344** | 6.2519 | 0.4203 | 1.3432 | 7.3879 | 15.6132 |
| Ours | 54.9875 | 0.6038 | **6.8965** | **0.4300** | **1.4038** | **7.4591** | **17.3038** |

**Table 4  Comparison of different attention modules on the TNO dataset.** The best values are shown in bold.

| Attention modules | EN | AG | VIF | MI | SF | SCD | CC |
|---|---|---|---|---|---|---|---|
| CBAM | 7.3959 | 7.1151 | 0.4728 | 2.0182 | **17.1461** | 1.4444 | 0.4281 |
| ECA | 7.3862 | 6.2146 | 0.5664 | 2.5768 | 15.6400 | **1.4918** | **0.4467** |
| SE (ours) | **7.4591** | **6.8965** | **0.6037** | **2.616**2 | 17.3038 | 1.4038 | 0.4300 |

## *Model component analysis*

In order to further verify the advantages of the SE module, we conducted an ablation study on it. We replace SE attention with CBAM module and ECA module, respectively, and compare the performance of these three methods. The experimental results are shown in Table 4. We found that the SE module performed best on several evaluation indicators. Specifically, SE modules outperformed CBAM and ECA modules in key indicators such as enhancement index (EN), average gradient (AG), visual information fidelity (VIF), mutual information (MI), spatial frequency (SF), and structural similarity (SCD). These results show that the SE module is able to capture and emphasize key features in the images more efficiently, thereby improving the quality of the fused images. So we finally chose the SE module in our network.

## FUTURE WORKS

For the experimental part of this study, we used the TNO image fusion dataset, which covers images in the enhanced vision (390–700 $nm$), near infrared (700–1,000 $nm$), and long wave infrared (8–12 $\mu m$) bands, covering a diverse range of military and surveillance scenarios, and showcases multiple targets, including people and vehicles. Although the TNO dataset provides a valuable resource for image fusion research, it also has some limitations that challenge the conclusions of this study and future work. The sample size of targets such as pedestrians in the TNO dataset is relatively small, and future studies will require a more diverse and rich sample of targets. At the same time, the images in the TNO dataset mostly show relatively simple scenes. Therefore, future research will focus on integrating larger and more diverse data sets to enhance the robustness and adaptability of the model. In summary, while the TNO dataset provides a valuable experimental basis for current research, we are also clearly aware of its limitations and will address these issues in future work with a view to advancing image fusion techniques.

In terms of the performance improvement of the model, we will conduct in-depth analysis on the performance of our proposed method in Qabf, SCD, and VIF indicators

and make improvements. We will consider enhancing the nonlinear fitting capability of our proposed method to mitigate spectral distortion to some extent. We will also explore modeling the requirements of high-level vision tasks into the entire image fusion process to further improve the performance of high-level vision tasks.

## CONCLUSION

In this article, an infrared and visible image fusion framework based on dense gradient attention residuals is proposed. The gradient attention residual-intensive module designed by the encoder can effectively extract strong and weak texture details. The SE network added to the module enhances the ability of the network to extract depth features. In addition, the introduced feature gradient attention module can further enhance the information extraction of the network and avoid the divergence of the infrared thermal radiation region. The energy fusion strategy of the fusion layer reduces the artifacts by assigning weights to the extracted features. We compare the proposed method with five commonly used methods on publicly available TNO data sets. The experimental results show that the proposed algorithm is superior in the similarity degree between the fusion image and the source image, the amount of information contained in the fusion image, and the visual effect of the fusion image. These results not only demonstrate the effectiveness of our approach but also demonstrate its potential for practical applications. In future work, we plan to validate our model under multiple data sets. At the same time, we further optimize our framework to improve its nonlinear fitting and other capabilities.

### Funding

This work was supported by the National Natural Science Foundation of China under Grant 61801319, the Sichuan Science and Technology Program under Grant 2020JDJQ0061 and Grant 2021YFG0099, the Innovation Fund of Chinese Universities under Grant 2020HYA04001, the Innovation Fund of Engineering Research Center of the Ministry of Education of China, Digital Learning Technology Integration and Application, under Grant 1221009, the 2022 Graduate Innovation Fund of Sichuan University of Science and Engineering under Grant Y2023297, the Engineering Research Center of Integration and Application of Digital Learning Technology, Ministry of Education, under Grant 1321002, and the Opening Project of Artificial Intelligence Key Laboratory of Sichuan Province under Grant 2021RZJ01. There was no additional external funding received for this study. The funders had no role in study design, data collection and analysis, decision to publish, or preparation of the manuscript.

### Grant Disclosures

The following grant information was disclosed by the authors:
National Natural Science Foundation of China: 61801319.
Sichuan Science and Technology Program: 2020JDJQ0061, 2021YFG0099.
Innovation Fund of Chinese Universities: 2020HYA04001.

Innovation Fund of Engineering Research Center of the Ministry of Education of China. Digital Learning Technology Integration and Application: 1221009.
2022 Graduate Innovation Fund of Sichuan University of Science and Engineering: Y2023297.
Engineering Research Center of Integration and Application of Digital Learning Technology, Ministry of Education: 1321002.
Opening Project of Artificial Intelligence Key Laboratory of Sichuan Province: 2021RZJ01.

## Competing Interests

The authors declare that they have no competing interests.

## Author Contributions

- Yongyu Luo conceived and designed the experiments, performed the experiments, analyzed the data, performed the computation work, prepared figures and/or tables, authored or reviewed drafts of the article, and approved the final draft.
- Zhongqiang Luo conceived and designed the experiments, performed the experiments, analyzed the data, performed the computation work, prepared figures and/or tables, authored or reviewed drafts of the article, and approved the final draft.

## Data Availability

The raw data and code for the experiment and the data sets used in the experiment are available in the Supplemental Files.

The TNO Image Fusion Dataset dataset is available at figshare: Toet, Alexander (2014). TNO Image Fusion Dataset. figshare. Dataset. https://doi.org/10.6084/m9.figshare.1008029.v2.

The Multi-Spectral dataset is available at GitHub: https://github.com/Linfeng-Tang/MSRS.

## Supplemental Information

Supplemental information for this article can be found online at http://dx.doi.org/10.7717/peerj-cs.2569#supplemental-information.

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
