# Peer review of "Infrared and visible image fusion algorithm based on gradient attention residuals dense block"

_PeerJ Computer Science, doi:10.7717/peerj-cs.2569_

## Round 0.1 · original submission · Major Revisions

Dear authors,
You are advised to critically respond to all comments point by point when preparing an updated version of the manuscript and while preparing for the rebuttal letter. Please address all comments/suggestions provided by reviewers, considering that these should be added to the new version of the manuscript.

Kind regards,
PCoelho

Reviewer 1 ·

Basic reporting

The manuscript is generally well-written with professional English. However, there are areas where clarity can be improved. For instance, some sections contain overly complex sentences, particularly between lines 101–105, which may confuse readers. Simplifying these sentences would enhance readability.

While the background section provides a decent overview, it lacks a detailed discussion of more recent advancements in the field of infrared-visible image fusion. Adding references to state-of-the-art methods from the past 2–3 years would give the research better context and showcase its relevance.

The article structure is clear, but the figures could be improved. Some figures are difficult to interpret due to inadequate labeling or contrast, particularly in Figures 6 and 7. Providing better labeling and enhancing the visual clarity of the figures would help in the interpretation of results.

Lastly, while the raw data is mentioned, a more detailed description of the dataset, especially regarding its limitations, is needed. Including these details would provide better insight into the scope and generalizability of the findings.

Experimental design

The experimental design is generally sound, but there are several areas that require improvement. First, the choice of the TNO dataset raises concerns about the diversity and relevance of the data used. This dataset is somewhat outdated and limited in scope, which may affect the generalizability of the results. It would be beneficial to include more recent and diverse datasets in the evaluation to demonstrate the robustness of the proposed method.

Additionally, while the ablation study is a positive inclusion, it lacks depth. The paper should provide a more comprehensive analysis of the components of the model, explaining why specific modules were selected over others and comparing their performance. This would give stronger justification for the design choices.

Furthermore, the experimental setup does not discuss computational efficiency, memory usage, or scalability, all of which are critical factors in real-world applications. Including these considerations would make the study more complete and applicable to practical scenarios. Finally, the comparison with state-of-the-art methods is insufficient. More detailed comparisons, particularly with methods published in the last 2–3 years, are necessary to properly assess the contribution of this work.

Validity of the findings

The findings presented in the manuscript show promising results in terms of traditional metrics like AG, EN, SF, MI, and SD. However, the validity of these findings is limited by the lack of comparison with more recent state-of-the-art methods. The absence of such comparisons makes it difficult to assess whether the proposed method truly offers significant improvements over existing approaches.

Furthermore, while the ablation study demonstrates some of the model’s performance, it does not provide sufficient justification for the chosen architecture. A deeper analysis is required to confirm the necessity of each component and whether simpler alternatives might achieve comparable results.

Additionally, the study lacks real-world evaluations. Although technical metrics are important, they do not fully capture the practical applicability of the method in real-world scenarios such as night vision or autonomous systems. More comprehensive evaluations, including qualitative assessments or real-time performance analysis, would strengthen the validity of the findings.

Finally, the lack of discussion regarding computational cost, scalability, and the method’s limitations leaves key questions about the practicality of deploying the proposed model unanswered. Addressing these aspects would make the conclusions more robust and applicable.

Additional comments

The proposed method introduces an interesting approach with gradient attention residuals, which could have potential in image fusion tasks. However, the manuscript would benefit from addressing the following points:

The method’s complexity should be justified by corresponding gains in performance. If the model introduces additional computational cost, this should be discussed, particularly regarding real-world applications.

A clearer comparison with state-of-the-art methods, both in terms of performance and computational efficiency, would strengthen the manuscript.

The inclusion of more recent datasets and broader testing conditions would enhance the relevance of the work and improve its generalizability.

By addressing these points, the paper will have a stronger foundation and demonstrate its practical relevance in the field of image fusion.

Reviewer 2 ·

Basic reporting

The paper introduces a novel method for fusing infrared and visible images to retain crucial texture and target information from both types of imagery. The authors propose integrating Squeeze-and-Excitation Networks into a gradient convolutional dense block, which improves the ability to extract important features while preserving details. An adaptive weighted energy attention network further enhances the fusion process. Authors claim that their method outperforms five common fusion methods on the TNO dataset, demonstrating superior performance in several metrics such as AG, EN, SF, MI, and SD.

Although the idea is valid, there are several problems that are pointed out below, and the authors need to answer them:

1. The abstract should be more focused on key points, and add some numerical results.

2. The introduction should be modified to effectively present the significance of the problem and a clearer statement of the contributions and novelty.

3. Highlight the specific gaps in current research that your method addresses and mention it in a bullet point format at the end of the Introduction section.

4. The Related Works section must be improved by adding more recent and cutting-edge research from 2023 and especially 2024. Some references are too old and not appropriate to utilize them.

Considering the following suggested papers to add them in Introduction or Related Works sections:

- Wang, Y., Pu, J., Miao, D., Zhang, L., Zhang, L., & Du, X. (2024). SCGRFuse: An infrared and visible image fusion network based on spatial/channel attention mechanism and gradient aggregation residual dense blocks. Engineering Applications of Artificial Intelligence, 132, 107898.

- Boroujeni, S. P. H., & Razi, A. (2024). Ic-gan: An improved conditional generative adversarial network for rgb-to-ir image translation with applications to forest fire monitoring. Expert Systems with Applications, 238, 121962.

- Huang, Z., Yang, B., & Liu, C. (2023). RDCa-Net: Residual dense channel attention symmetric network for infrared and visible image fusion. Infrared Physics & Technology, 130, 104589.

5. Modify the Related Works section by discussing the limitations of existing approaches explicitly to better present the novel contributions of your work. Add the pros and cons of each method.

6. Suggest specific directions for future research, including potential improvements to your method.

7. There are some grammatical errors in the paper. Check the paper for grammatical errors and ensure it follows a consistent academic writing style.

Experimental design

8. Expand the justification for choosing the five specific methods used for comparison (PMGI, DenseFuse, FusionGAN, etc.). Including more recent or diverse fusion techniques to provide a more comprehensive comparison.

9. The authors mention using batch size, epochs, and learning rate, but providing a reason behind these choices and their impact on convergence and model stability could enhance the reader's understanding.

10. Address the computational complexity of the proposed method in comparison to the other methods. Providing details on the training time, memory usage, and overall resource requirements could be helpful for practical deployment.

Validity of the findings

11. Consider including additional image quality metrics (e.g., PSNR, SSIM) to complement the current metrics and offer a more comprehensive evaluation of the fusion results, especially in terms of image quality and perceptual significance.

12. Add statistical significance tests for the comparative results to validate the superiority of the proposed method over other methods and ensure that improvements are not due to random variance.

---

## Round 0.2 · accepted · Accept

Dear authors, we are pleased to verify that you meet the reviewer's valuable feedback to improve your research.

Thank you for considering PeerJ Computer Science and submitting your work.

Reviewer 1 ·

Basic reporting

The article may be accepted in its current form.

Experimental design

The article may be accepted in its current form.

Validity of the findings

The article may be accepted in its current form.

Additional comments

The article may be accepted in its current form.

Reviewer 2 ·

Basic reporting

The authors have completely addressed all my comments, and I have no further concerns. Therefore, I recommend accepting the paper.

Experimental design

The authors have completely addressed all my comments, and I have no further concerns. Therefore, I recommend accepting the paper.

Validity of the findings

The authors have completely addressed all my comments, and I have no further concerns. Therefore, I recommend accepting the paper.